# Efficient Generation of Structured Objects with Constrained Adversarial Networks

**Luca Di Liello**[*]
University of Trento

**Pierfrancesco Ardino**[*]
University of Trento

**Jacopo Gobbi**[*]
University of Trento

**Paolo Morettin**[†]
KU Leuven

**Stefano Teso**
University of Trento
`firstname.lastname@unitn.it`

**Andrea Passerini**
University of Trento

## Abstract

Generative Adversarial Networks (GANs) struggle to generate structured objects like molecules and game maps. The issue is that structured objects must satisfy hard requirements (e.g., molecules must be chemically valid) that are difficult to acquire from examples alone. As a remedy, we propose Constrained Adversarial Networks (CANs), an extension of GANs in which the constraints are embedded into the model during training. This is achieved by penalizing the generator proportionally to the mass it allocates to invalid structures. In contrast to other generative models, CANs support efficient inference of valid structures (with high probability) and allows to turn on and off the learned constraints at inference time. CANs handle arbitrary logical constraints and leverage knowledge compilation techniques to efficiently evaluate the disagreement between the model and the constraints. Our setup is further extended to hybrid logical-neural constraints for capturing very complex constraints, like graph reachability. An extensive empirical analysis shows that CANs efficiently generate valid structures that are both high-quality and novel.

## 1 Introduction

Many key applications require to generate objects that satisfy hard structural constraints, like drug molecules, which must be chemically valid, and game levels, which must be playable. Despite their impressive success [1, 2, 3], Generative Adversarial Networks (GANs) [4] struggle in these applications. The reason is that data alone are often insufficient to capture the structural constraints (especially if noisy) and convey them to the model.

As a remedy, we derive Constrained Adversarial Networks (CANs), which extend GANs to generating valid structures with high probabilty. Given a set of arbitrary discrete constraints, CANs achieve this by penalizing the generator for allocating mass to invalid objects during training. The penalty term is implemented using the semantic loss (SL) [5], which turns the discrete constraints into a differentiable loss function implemented as an arithmetic circuit (i.e., a polynomial). The SL is probabilistically sound, can be evaluated exactly, and supports end-to-end training. Importantly, the polynomial – which can be quite large, depending on the complexity of the constraints – can be thrown away after training. In addition, CANs handle complex constraints, like reachability on graphs, by first embedding the candidate configurations in a space in which the constraints can be encoded compactly, and then applying the SL to the embeddings.

---

[*]Equal contributions.
[†]This work was partially carried out when PM was working at the University of Trento.

Since the constraints are embedded directly into the generator, high-quality structures can be sampled efficiently (in time practically independent of the complexity of the constraints) with a simple forward pass on the generator, as in regular GANs.[3] No costly sampling or optimization steps are needed. We additionally show how to equip CANs with the ability to switch constraints on and off dynamically during inference, at no run-time cost.

**Contributions.** Summarizing, we contribute: 1) CANs, an extension of GANs in which the generator is encouraged at training time to generate valid structures and support efficient sampling, 2) native support for intractably complex constraints, 3) conditional CANs, an effective solution for dynamically turning on and off the constraints at inference time, 4) a thorough empirical study on real-world data showing that CANs generate structures that are likely valid and coherent with the training data.

## 2   Related Work

Structured generative tasks have traditionally been tackled using probabilistic graphical models [7] and grammars [8], which lack support for representation learning and efficient sampling under constraints. Tractable probabilistic circuits [9, 10] are a recent alternative that make use of ideas from knowledge compilation [11] to provide efficient generation of valid structures. These approaches generate valid objects by constructing a circuit (a polynomial) that encodes both the hard constraints and the probabilistic structure of the problem. Although inference is linear in the size of the circuit, the latter can grow very large if the constraints are complex enough. In contrast, CANs model the probabilistic structure of the problem using a neural architecture, while relying on knowledge compilation for encoding the hard constraints during training. Moreover, the circuit can be discarded at inference time. The time and space complexity of sampling for CANs is therefore roughly independent from the complexity of the constraints in practice.

Deep generative models developed for structured tasks are special-purpose, in that they rely on ad-hoc architectures, tackle specific applications, or have no support for efficient sampling [12, 13, 14, 15]. Some recent approaches have focused on incorporating a constraint learning component in training deep generative models, using reinforcement learning [13] or inverse reinforcement learning [16] techniques. This direction is complementary to ours and is useful when constraints are not known in advance or cannot be easily formalized as functions of the generator output. Indeed, our experiment on molecule generation shows the advantages of enriching CANs with constraint learning to generate high quality and diverse molecules.

Other general approaches for injecting knowledge into neural nets (like deep statistical-relational models [17, 18, 19], tensor-based models [20, 21], and fuzzy logic-based models [22]) are either not generative or require the constraints to be available at inference time.

## 3   Unconstrained GANs

GANs [4] are composed of two neural nets: a discriminator $d$ trained to recognize "real" objects $\mathbf{x} \in \mathcal{X}$ sampled from the data distribution $P_r$, and a generator $g : \mathcal{Z} \to \mathcal{X}$ that maps random latent vectors $\mathbf{z} \in \mathcal{Z}$ to objects $g(\mathbf{x})$ that fool the discriminator. Learning equates to solving the minimax game $\min_g \max_d \ f_{\text{GAN}}(g, d)$ with value function:

$$f_{\text{GAN}}(g, d) := \mathbb{E}_{\mathbf{x} \sim P_r}[\log P_d(\mathbf{x})] + \mathbb{E}_{\mathbf{x} \sim P_g}[\log(1 - P_d(\mathbf{x}))] \qquad (1)$$

Here $P_g(\mathbf{x})$ and $P_d(\mathbf{x}) := P_d(\text{real} \mid \mathbf{x})$ are the distributions induced by the generator and discriminator, respectively. New objects $\mathbf{x}$ can be sampled by mapping random vectors $\mathbf{z}$ using the generator, i.e., $\mathbf{x} = g(\mathbf{z})$. Under idealized assumptions, the learned generator matches the data distribution:

**Theorem 1** ([4]). *If $g$ and $d$ are non-parametric and the leftmost expectation in Eq. 1 is approximated arbitrarily well by the data, the global equilibrium $(g^*, d^*)$ of Eq. 1 satisfies $P_{d^*} \equiv \frac{1}{2}$ and $P_{g^*} \equiv P_r$.*

In practice, training GANs is notoriously hard [23, 24]. The most common failure mode is mode collapse, in which the generated objects are clustered in a tiny region of the object space. Remedies include using alternative objective functions [4], divergences [25, 26] and regularizers [27]. In our experiments, we apply some of these techniques to stabilize training.

In structured tasks, the objects of interest are usually discrete. In the following, we focus on stochastic generators that output a *categorical distribution* $\boldsymbol{\theta}(\mathbf{z})$ over $\mathcal{X}$ and objects are sampled from the latter. In this case, $P_g(\mathbf{x}) = \int_{\mathcal{Z}} P_g(\mathbf{x}|\mathbf{z})p(\mathbf{z})d\mathbf{z} = \int_{\mathcal{Z}} \boldsymbol{\theta}(\mathbf{z})p(\mathbf{z})d\mathbf{z} = \mathbb{E}_{\mathbf{z}}[\boldsymbol{\theta}(\mathbf{z})]$.

# 4 Generating Structures with CANs

Our goal is to learn a deep generative model that outputs structures $\mathbf{x}$ consistent with validity constraints and an unobserved distribution $P_r$. We assume to be given: i) a feature map $\phi : \mathcal{X} \to \{0, 1\}^b$ that extracts $b$ binary features from $\mathbf{x}$, and ii) a single validity constraint $\psi$ encoded as a Boolean formula on $\phi(\mathbf{x})$. If $\mathbf{x}$ is binary, $\phi$ can be taken to be the identity; later we will discuss some alternatives. Any discrete structured space can be encoded this way.

## 4.1 Limitations of GANs

Standard GANs struggle to output valid structures, for two main reasons. First, the number of examples necessary to capture any non-trivial constraint $\psi$ can be intractably large.[4] This rules out learning the rules of chemical validity or, worse still, graph reachability from even moderately large data sets. Second, in many cases of interest the examples are noisy and do violate $\psi$, in which case the data lures GANs into learning *not* to satisfy the constraint:

**Corollary 1.** *Under the assumptions of Theorem 1, given a target distribution $P_r$, a constraint $\psi$ consistent with it, and a dataset of examples $\mathbf{x}$ sampled i.i.d. from a corrupted distribution $P_r' \neq P_r$ inconsistent with $\psi$, GANs associate non-zero mass to infeasible objects.*

This follows easily from Theorem 1, as the optimal generator satisfies $P_g \equiv P_r'$, which is inconsistent with $\psi$. Since Theorem 1 captures the *intent* of GAN training, this corollary shows that GANs are *by design* incapable of handling invalid examples.

## 4.2 Constrained Adversarial Networks

Constrained Adversarial Networks (CANs) avoid these issues by taking both the data and the target structural constraint $\psi$ as inputs. The value function is designed so that the generator maximizes the probability of generating valid structures. In order to derive CANs it is convenient to start from the following alternative GAN value function [4]: $f_{\text{ALT}}(g, d) := \mathbb{E}_{\mathbf{x} \sim P_r}[\log P_d(\mathbf{x})] - \mathbb{E}_{\mathbf{x} \sim P_g}[\log P_d(\mathbf{x})]$.

Let $(g, d)$ be a GAN and $v(\mathbf{x}) = \mathbb{1}\{\phi(\mathbf{x}) \models \psi\}$ be a fixed discriminator that distinguishes between valid and invalid structures, where $\models$ indicates logical entailment. Ideally, we wish the generator to *never* output invalid structures. This can be achieved by using an aggregate discriminator $a(\mathbf{x})$ that only accepts configurations that are both valid and high-quality w.r.t. $d$. Let $A$ be the indicator that $a$ classifies $\mathbf{x}$ as real, and similarly for $D$ and $V$. By definition:

$$P_a(\mathbf{x}) = P(A \,|\, \mathbf{x}) = P(D \,|\, V, \mathbf{x})P(V \,|\, \mathbf{x}) = P_d(\mathbf{x})\mathbb{1}\{\phi(\mathbf{x}) \models \psi\} \tag{2}$$

Plugging the aggregate discriminator into the alternative value function gives:

$$\arg\max_{a} \; f_{\text{ALT}}(g, a) \tag{3}$$

$$= \arg\max_{d} \; \mathbb{E}_{P_r}[\log P_d(\mathbf{x}) + \log \mathbb{1}\{\phi(\mathbf{x}) \models \psi\}] - \mathbb{E}_{P_g}[\log P_d(\mathbf{x}) + \log \mathbb{1}\{\phi(\mathbf{x}) \models \psi\}] \tag{4}$$

$$= \arg\max_{d} \; \mathbb{E}_{P_r}[\log P_d(\mathbf{x})] - \mathbb{E}_{P_g}[\log P_d(\mathbf{x})] - \mathbb{E}_{P_g}[\log \mathbb{1}\{\phi(\mathbf{x}) \models \psi\}] \tag{5}$$

$$= \arg\max_{d} \; f_{\text{ALT}}(g, d) - \mathbb{E}_{P_g}[\log \mathbb{1}\{\phi(\mathbf{x}) \models \psi\}] \tag{6}$$

The second step holds because $\mathbb{E}_{P_r}[\log \mathbb{1}\{\phi(\mathbf{x}) \models \psi\}]$ does not depend on $d$. If $g$ allocates non-zero mass to *any* measurable subset of invalid structures, the second term becomes $+\infty$. This is consistent with our goal but problematic for learning. A better alternative is to optimize the lower bound:

$$SL_{\psi}(g) := -\log P_g(\psi) = -\log \mathbb{E}_{P_g}[\mathbb{1}\{\phi(\mathbf{x}) \models \psi\}] \leq -\mathbb{E}_{P_g}[\log \mathbb{1}\{\phi(\mathbf{x}) \models \psi\}] \tag{7}$$

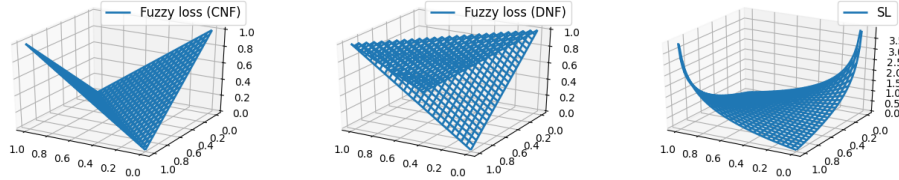

Figure 1: Left: fuzzy logic encoding (using the Łukasiewicz T-norm) of $x \oplus y$ in CNF format as a function of $P(x = 1)$ and $P(y = 1)$. Middle: encoding of DNF XOR. Right: SL of either encoding.

This term is the *semantic loss* (SL) proposed in [5] to inject knowledge into neural networks. The SL is much smoother than the original and it only evaluates to $+\infty$ if $P_g$ allocates *all* the mass to infeasible configurations. This immediately leads to the CAN value function:

$$f_{\text{CAN}}(g, d) := f_{\text{ALT}}(g, d) + \lambda SL_\psi(g) \tag{8}$$

where $\lambda > 0$ is a hyper-parameter controlling the importance of the constraint. This formulation is related to integral probability metric-based GANs, cf. [29]. The SL can be viewed as the negative log-likelihood of $\psi$, and hence it rewards the generator proportionally to the mass it allocates to valid structures. The expectation in Eq. 7 can be rewritten as:

$$\mathbb{E}_{\mathbf{x} \sim P_g}[\mathbb{1}\{\phi(\mathbf{x}) \models \psi\}] = \sum_{\mathbf{x} : \phi(\mathbf{x}) \models \psi} P_g(\mathbf{x}) = \mathbb{E}_{\mathbf{z}} \left[ \sum_{\mathbf{x} : \phi(\mathbf{x}) \models \psi} \prod_{i \, : \, x_i = 1} \theta_i(\mathbf{z}) \prod_{i \, : \, x_i = 0} (1 - \theta_i(\mathbf{z})) \right] \tag{9}$$

Hence, the SL is the negative logarithm of a polynomial in $\boldsymbol{\theta}$ and it is fully differentiable.[5] In practice, below we apply the semantic loss term directly to $f_{\text{GAN}}$, i.e., $f_{\text{CAN}}(g, d) := f_{\text{GAN}}(g, d) + \lambda SL_\psi(g)$.

If the SL is given large enough weight $\lambda$ then it gets closer to the ideal "hard" discriminator, and therefore more strongly encourages the CAN to generate valid structures. Under the preconditions of Theorem 1, it is clear that for $\lambda \to \infty$ CANs generate valid structures only:

**Proposition 1.** *Under the assumptions of Corollary 1, CANs associate zero mass to infeasible objects, irrespective of the discrepancy between $P_r$ and $P_r{}'$.*

Indeed, any global equilibrium $(g^*, d^*)$ of $\min_g \max_d f_{\text{CAN}}(g, d)$ minimizes the second term: the minimum is attained by $\log P_{g^*}(\psi) = 0$, which entails $P_{g^*}(\neg\psi) = 0$. Of course, as with standard GANs, the prerequisites are often violated in practice. Regardless, Proposition 1 works as a sanity check, and shows that, in contrast to GANs, CANs are appropriate for structured generative tasks.

A possible alternative to the SL is to introduce a differentiable knowledge-based loss into the value function by relaxing the constraint $\psi$ using fuzzy logic, as done in a number of recent works on discriminative deep learning [21, 22]. Apart from lacking a formal derivation in terms of expected probability of satisfying constraints, the issue is that fuzzy logic is not semantically sound, meaning that equivalent encodings of the same constraint may give different loss functions [30]. Figure 1 illustrates this on an XOR constraint: the "fuzzy loss" and its gradient change radically depending on whether the XOR is encoded as CNF (left) or DNF (middle), while the SL is unaffected (right).

**Evaluating the Semantic Loss** The sum in Eq. 9 is the unnormalized probability of generating a valid configuration. Evaluating it requires to sum over *all* solutions of $\psi$, weighted according to their probability with respect to $\boldsymbol{\theta}$. This task is denoted Weighted Model Counting (WMC) [31]. Naïvely implementing WMC is infeasible in most cases, as it involves summing over exponentially many configurations. Knowledge compilation (KC) [11] is a well known approach in automated reasoning and solving WMC through KC is the state-of-the-art for answering probabilistic queries in discrete graphical models [31, 32, 33]. Roughly speaking, KC leverages distributivity to rewrite the polynomial in Eq. 9 as compactly as possible, often offering a tremendous speed-up during evaluation. This is achieved by identifying shared sub-components and compactly representing the factorized polynomial using a DAG. Target representations for the DAG (OBDDs, DNNFs, *etc.* [11]) differ in succinctness and enable different polytime (in the size of the DAG) operations. As done in [5], we

compile the SL polynomial into a Sentential Decision Diagram (SDD) [34] that enables efficient WMC and therefore exact evaluation of the SL and of its gradient. KC is key in making evaluation of the Semantic Loss and of its gradient practical, at the cost of an offline compilation step – which is however performed only once before training.

The main downside of KC is that, depending on the complexity of $\psi$, the compiled circuit may be large. This is less of an issue during training, which is often performed on powerful machines, but it can be problematic for inference, especially on embedded devices. A major advantage of CANs is that the circuit is not required for inference (as the latter consists of a simple forward pass over the generator), and can thus be thrown away after training. This means that CANs incur no space penalty during inference compared to GANs.

**The embedding function $\phi$** The embedding function $\phi(\mathbf{x})$ extracts Boolean variables to which the SL is then applied. In many cases, as in our molecule experiment, $\phi$ is simply the identity map. However, when fed a particularly complex constraint $\psi$, KC my output an SDD too large even for the training stage. In this case, we use $\phi$ to map $\mathbf{x}$ to an application-specific embedding space where $\psi$ (and hence the SL polynomial) is expressible in compact form. We successfully employed this technique to synthesize Mario levels where the goal tile is reachable from the starting tile; all details are provided below. The same technique can be exploited for dealing with other complex logical formulas beyond the reach of state-of-the-art knowledge compilation.

### 4.3 Conditional CANs

So far we described how to use the SL for enforcing structural constraints on the generator's output. Since the SL can be applied to any distribution over binary variables, it can also be used to enforce conditional constraints that can be turned on and off at inference time. Specifically, we notice that the constraint can involve also latent variables, and we show how this can be leveraged for different purposes. Similarly to InfoGANs [35], the generator's input is augmented with an additional binary vector $\mathbf{c}$. Instead of maximizing (an approximation of) the mutual information between $\mathbf{c}$ and the generator's output, the SL is used to logically bind the input codes to semantic features or constraint of interest. Let $\psi_1, \ldots, \psi_k$ be $k$ constraints of interest. In order to make them switchable, we extend the latent vector $\mathbf{z}$ with $k$ fresh variables $\mathbf{c} = (c_1, \ldots, c_k) \in \{0,1\}^k$ and train the CAN using the constraint:

$$\psi = \bigwedge_{i=1}^{k} (c_i \leftrightarrow \psi_i)$$

where the prior $P(\mathbf{c})$ used during training is estimated from data.

Using a conditional SL term during training results in a model that can be conditioned to generate object with desired, arbitrarily complex properties $\psi_i$ at inference time. Additionally, this feature shows a beneficial effect in mitigating mode collapse during training, as reported in Section 5.2.

## 5 Experiments

Our experimental evaluation aims at answering the following questions:

**Q1** Can CANs with tractable constraints achieve better results than GANs?

**Q2** Can CANs with intractable constraints achieve better results than GANs?

**Q3** Can constraints be combined with rewards to achieve better results than using rewards only?

We implemented CANs[6] using Tensorflow and used PySDD[7] to perform knowledge compilation. We tested CANs using different generator architectures on three real-world structured generative tasks.[8] In all cases, we evaluated the objects generated by CANs and those of the baselines using three metrics (adopted from [36]): **validity** is the proportion of sampled objects that are valid; **novelty** is the proportion of valid sampled objects that are not present in the training data; and **uniqueness** is the proportion of valid unique (non-repeated) sampled objects.

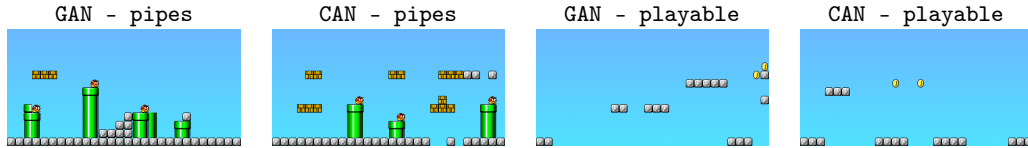

Figure 2: Examples of SMB levels generated by GAN and CAN. Left: generating levels containing pipes; right: generating reachable levels. For each of the two settings we report prototypical examples of levels generated by GAN (first and third picture) and CAN (second and fourth picture). Notice how all pipes generated by CAN are valid, contrarily to what happens for GAN, and that the GAN generates a level that is not playable (because of the big jump at the start of the map).

## 5.1 Super Mario Bros level generation

In this experiment we show how CANs can help in the challenging task of learning to generate videogame levels from user-authored content. While procedural approaches to videogame level generation have successfully been used for decades, the application of machine learning techniques in the creation of (functional) content is a relatively new area of research [37]. On the one hand, modern video game levels are characterized by aesthetical features that cannot be formally encoded and thus are difficult to implement in a procedure, which motivates the use of ML techniques for the task. On the other hand, the levels have often to satisfy a set of functional (hard) constraints that are easy to guarantee when the generator is hand-coded but pose challenges for current machine learning models.

Architectures for Super Mario Bros level generation include LSTMs [38], probabilistic graphical models [39], and multi-dimensional MCMC [40]. MarioGANs [15] are specifically designed for level generation, but they only constrain the mixture of tiles appearing in the level. This technique cannot be easily generalized to arbitrary constraints.

In the following, we show how the semantic loss can be used to encode useful hard constraints in the context of videogame level generation. These constraints might be functional requirements that apply to every generated object or might be contextually used to steer the generation towards objects with certain properties. In our empirical analysis, we focus on *Super Mario Bros* (SMB), possibly one of the most studied video games in tile-based level generation.

Recently, [41] applied Wasserstein GANs (WGANs) [26] to SMB level generation. The approach works by first training a generator in the usual way, then using an evolutionary algorithm called Covariance Matrix Adaptation Evolution Strategy (CMA-ES) to search for the best latent vectors according to a user-defined fitness function on the corresponding levels. We stress that this technique is orthogonal to CANs and the two can be combined together. We adopt the same experimental setting, WGAN architecture and training procedure of [41]. The structured objects are $14 \times 28$ tile-based representations of SMB levels (e.g. Fig. 2) and the training data is obtained by sliding a $28$ tiles window over levels from the *Video game level corpus* [42].

We run all the experiments on a machine with a single 1080Ti GPU for $4$ times with random seeds.

### 5.1.1 CANs with tractable constraints: generating SMB levels with *pipes*

In this experiment, the focus is on showing how CANs can effectively deal with constraints that can be directly encoded over the generator output. Pipes are made of four different types of tiles. They can have a variable height but the general structure is always the same: two tiles (*top-left* and *top-right*) on top and one or more pairs of body tiles (*body-left* and *body-right*) below (see the `CAN -` `pipes` in picture in Fig. 2 for examples of valid pipes). Since encoding all possible dispositions and combinations of pipes in a level would result in an extremely large propositional formula, we apply the constraint locally to a $2 \times 2$ window that is slid, horizontally and vertically, by one tile at a time (notice that all structural properties of pipes are covered using this method). The constraint consists of a lot of implications of the type "if this is a *top-left* tile, then the tile below must be a *body-left* one" conjoined together (see the Supplementary material for the full formula). The relative importance of the constraints is determined by the hyper-parameter $\lambda$ (see Eq. 8).

There are two major problems in the application of the constraint on pipes when using a large $\lambda$: i) *vanishing pipes*: this occurs because the generator can satisfy the constraint by simply generating

| Model | # Maps | Validity (%) | Average pipe-tiles / level | L1 Norm | Training time |
|-------|--------|--------------|----------------------------|---------|---------------|
| GAN | 7 | 47.6 ± 8.3 | 7.8 | 0.0115 | 1h 12m |
| CAN | 7 | 83.2 ± 4.8 | 6.4 | 0.0110 | 2h 2m |

Table 1: Comparison between GAN and CAN on SMB level generation with pipes. The 7 maps containing pipes are *mario-1-1*, *mario-2-1*, *mario-3-1*, *mario-4-1*, *mario-4-2*, *mario-6-2* and *mario-8-1*, for a total of $1,404$ training samples. Results report validity, average number of pipe tiles per level, L1 norm on the difference between each pair of levels in the generated batch and training time. Inference is real-time (< 40 ms) for both architectures.

| Network type | Level | Tested samples | Validity | Training time | Inference time per sample |
|--------------|-------|----------------|----------|---------------|---------------------------|
| GAN | mario-1-3 | 1000 | 9.80% | 1 h 15 min | ∼ 40 ms |
| GAN + CMA-ES | mario-1-3 | 1000 | 65.90% | 1 h 15 min | ∼ 22 min |
| CAN | mario-1-3 | 1000 | 71.60% | 1 h 34 min | ∼ 40 ms |
| GAN | mario-3-3 | 1000 | 13.00% | 1 h 11 min | ∼ 40 ms |
| GAN + CMA-ES | mario-3-3 | 1000 | 64.20% | 1 h 11 min | ∼ 22 min |
| CAN | mario-3-3 | 1000 | 62.30% | 1 h 27 min | ∼ 40 ms |

Table 2: Results on the generation of *playable* SMB level. Levels *mario-1-3* (123 training samples) and *mario-3-3* (122 training samples) were chosen due to their high solving complexity. Results compare a baseline GAN, a GAN combined with CMA-ES and a CAN. Validity is defined as the ability of the A* agent to complete the level. Note that inference time for GAN and CAN is measured in milliseconds while time for GAN + CMA-ES is in minutes.

layers without pipes; ii) *mode collapse*: the generator may learn to place pipes always in the same positions. We address both issues by introducing the SL after an initial bootstrap phase (of $5,000$ epochs) in which the generator learns to generate sensible objects, and by linearly increasing its weight from zero to $\lambda = 0.2$. The final value for $\lambda$ was chosen as the highest value allowing to retain al least 80% of pipe tiles on average with respect to a plain GAN. All experiments were run for $12,000$ epochs.

Table 1 reports experimental results comparing GAN and CAN trained on all levels containing pipes. CAN manage to almost double the validity of the generated levels (see the two left pictures in Fig. 2 for some prototypical examples) while retaining about 82% of the pipe tiles and without any significant loss in terms of diversity (as measured by the L1 norm on the difference between each pair of levels in the generated batch) or cost in terms of training (roughly doubled training times). Inference is real-time (< 40 ms) for both architectures.

These results allow to answer **Q1** affirmatively.

### 5.1.2 CANs with intractable constraints: generating *playable* SMB levels

In the following we show how CANs can be successfully applied in settings where constraints are too complex to be directly encoded onto the generator output. A level is *playable* if there is a feasible path[9] from the left-most to the right-most column of the level. We refer to this property as *reachability*. We compare CANs with CMA-ES, as both techniques can be used to steer the network towards the generation of playable levels. In CMA-ES, the fitness function doesn't have to be differentiable and the playability is computed on the output of an A* agent (the same used in [41]) playing the level. Having the SL to steer the generation towards playable levels is not trivial, since it requires a differentiable definition of playability. Directly encoding the constraint in propositional logic is intractable. Consider the size of a first order logic propositional formula describing all possible path a player can follow in the level. We thus define the playability constraint on the output of an embedding function $\phi$ (modelled as a feedforward NN) that approximates tile reachability. The function is trained to predict whether each tile is reachable from the left-most column using traces obtained from the A* agent. See the Supplementary material for the details.

Table 2 shows the validity of a batch of $1,000$ levels generated respectively by plain GAN, GAN combined with CMA-ES using the default parameters for the search, and a forward pass of CAN. Each training run lasted 15000 epochs with all the default hyper parameters defined in [41], and the SL was activated from epoch 5000 with $\lambda = 0.01$, which validation experiments showed to

| Reward for | SL | validity | uniqueness | diversity | QED | SA | logP |
|---|---|---|---|---|---|---|---|
| QED + SA + logP | False | 97.4 | 2.4 | 91.0 | 47.0 | 84.0 | 65.0 |
| | True | 96.6 | 2.5 | 98.8 | 51.8 | 90.7 | 73.6 |

Table 3: Results of using the semantic loss on the MolGAN architecture. The diversity score is obtained by comparing sub-structures of generated samples against a random subset of the dataset. A lower score indicates a higher amount of repetitions between the generated samples and the dataset. The first row refers to the results reported in the MolGAN paper.

be a reasonable trade-off between SL and generator loss. Results show that CANs achieves better (*mario-1-3*) or comparable (*mario-3-3*) validity with respect to GAN + CMA-ES at a fraction of the inference time. At the cost of pretraining the reachability function, CANs avoid the execution of the A* agent during the generation and sample high quality objects in milliseconds (as compared to minutes), thus enabling applications to create new levels at run time. Moreover, no significant quality degradation can be seen on the generated levels as compared to the ones generated by plain GAN (which on the other hand fails most of the time to generate reachable levels), as can be seen in Fig. 2. With these results, we can answer **Q2** affirmatively.

## 5.2 Molecule generation

Most approaches in molecule generation use variational autoencoders (VAEs) [43, 44, 45, 46], or more expensive techniques like MCMC [47]. Closest to CANs are ORGANs [12] and MolGANs [13], which respectively combine Sequence GANs (SeqGANs) and Graph Convolutional Networks (GCNs) with a reward network that optimizes specific chemical properties. Albeit comparing favorably with both sequence models [48, 12] (using SMILE representations) and likelihood-based methods, MolGAN are reported to be susceptible to mode collapse.

In this experiment, we investigate **Q3** by combining MolGAN's adversarial training and *reinforcement learning objective* with a conditional SL term on the task of generating molecules with certain desirable chemical properties. In contrast with our previous experimental settings, here the structured objects are undirected graphs of bounded maximum size, represented by discrete tensors that encode the atom/node type (padding atom (no atom), Carbon, Nitogren, Oxygen, Fluorine) and the bound/edge type (padding bond (no bond), single, double, triple and aromatic bond). During training, the network implicitly rewards validity and the maximization of the three chemical properties at once: **QED** (druglikeness), **SA** (synthesizability) and **logP** (solubility). The training is stopped once the uniqueness drops under $0.2$. We augment the MolGAN architecture with a conditional SL term, making use of $4$ latent dimensions to control the presence of one of the $4$ types of atoms considered in the experiment, as shown in Section 4.3.

Conditioning the generation of molecules with specific atoms at training time mitigates the drop in uniqueness caused by the reward network during the training. This allows the model to be trained for more epochs and results in more diverse and higher quality molecules, as reported in Table 3.

In this experiment, we train the model on a NVIDIA RTX 2080 Ti. The total training time is around 1 hour, and the inference is real-time. Using CANs produced a negligible overhead during the training with respect to the original model, providing further evidence that the technique doesn't heavily impact on the training. This results suggest that coupling CANs with a reinforcement learning objective is beneficial, answering **Q3** affirmatively.

## 6 Conclusion

We presented Constrained Adversarial Networks (CANs), a generalization of GANs in which the generator is encouraged *during training* to output valid structures. CANs make use of the semantic loss [5] to penalize the generator proportionally to the mass it allocate to invalid structures and. As in GANs, generating valid structures (on average) requires a simple forward pass on the generator. Importantly, the data structures used by the SL, which can be large if the structural constraints are very complex, are discarded after training. CANs were proven to be effective in improving the quality of the generated structures without significantly affecting inference run-time, and conditional CANs proved useful in promoting diversity of the generator's outputs.

## Broader Impact

Broadly speaking, this work aims at improving the reliability of structures / configurations generated via machine learning approaches. This can have a strong impact on a wide range of research fields and application domains, from drug design and protein engineering to layout synthesis and urban planning. Indeed, the lack of reliability of machine-generated outcomes is one of main obstacles to a wider adoption of machine learning technology in our societies. On the other hand, there is a risk of overestimating the reliability of the outputs of CANs, which are only guaranteed to satisfy constraints in expectation. For applications in which invalid structures should be avoided, like safety-critical applications, the objects output by CANs should always be validated before use.

From an artificial intelligence perspective, this work supports the line of thought that in order to overcome the current limitations of AI there is a need for combining machine learning and especially deep learning technology with approaches from knowledge representation and automated reasoning, and that principled ways to achieve this integration should be pursued.

## Acknowledgments and Disclosure of Funding

We are grateful to Gianvito Taneburgo for carrying out some preliminary experiments and to the anonymous reviewers for helping to improve the manuscript. This work has received funding from the European Research Council (ERC) under the European Union's Horizon 2020 research and innovation programme (grant agreement No. [694980] SYNTH: Synthesising Inductive Data Models). The research of ST has received funding from the "DELPhi - DiscovEring Life Patterns" project funded by the MIUR Progetti di Ricerca di Rilevante Interesse Nazionale (PRIN) 2017 – DD n. 1062 del 31.05.2019. The research of AP was partially supported by TAILOR, a project funded by EU Horizon 2020 research and innovation programme under GA No 952215.

## Footnotes

[3] With high probability. Invalid structures, when generated, can be checked and rejected efficiently. In this sense, CANs are related to learning efficient proposal distributions [6].

[4]The VC dimension of unrestricted discrete formulas is exponential in the number of variables [28].

[5]As long as $P_g(\mathbf{x}) > 0$, which is always the case in practice.

[6]The code is freely available at https://github.com/unitn-sml/CAN

[7]URL: `pypi.org/project/PySDD/`

[8]Details can be found in the Supplementary material.

[9]According to the game's physics.

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
