[Supplementary Material]

# Supplementary Material for Efficient Generation of Structured Objects with Constrained Adversarial Networks

**Luca Di Liello**[*]
University of Trento

**Pierfrancesco Ardino**[*]
University of Trento

**Jacopo Gobbi**[*]
University of Trento

**Paolo Morettin**[†]
KU Leuven

**Stefano Teso**
University of Trento
firstname.lastname@unitn.it

**Andrea Passerini**
University of Trento

## 1 Implementation details

### 1.1 Super Mario Bros Level Generation

The deep neural network for this experiment is based on the DCGANs used in [1]. Batch normalization and ReLU are applied between the layers of the generator $g$, while batch normalization and Leaky ReLU with a slope of $0.2$ has been used for the discriminator $d$. In the last layer of the generator we apply a *softmax* activation function to obtain probabilities that are finally given in input to the Semantic Loss. On the other hand, the generation of samples is done through the application, always on $g$, of a stretched softmax function followed by an *argmax*, as in [1].

The networks have been trained using the WGAN guidelines [2]. Thus, the number of iterations on the discriminator has been set to 5 for each iteration on the generator. *RMSProp* has been used as optimizer, with a constant learning rate equal to $0.00005$. The batch size used during the experiments has been set to 32. Layers have been initialized using *normal* initializer for both the generator and the discriminator. Moreover, weight clipping is applied on the weights of $d$ with $c$ equal to $0.01$. Finally, the size of the latent vector has been set to 32 and sampled from a normal distribution $\mathcal{N}(\mathbf{0}, \mathbf{1})$ Table 1 shows the network architecture of $g$ and $d$. In experiments where the SL is enabled only after some epoch $e$, in the first epochs before $e$ only GANs runs are executed. After that threshold, CAN runs are created as a fork of GAN runs.

| Part | Input Shape → Output Shape | Layer Type | Kernel | Stride |
|---|---|---|---|---|
| | $(32) \to (1, 1, 32)$ | Reshape. | - | - |
| | $(1, 1, 32) \to (\frac{h}{8}, \frac{w}{8}, 16)$ | Deconv. | $4 \times 4$ | $1 \times 1$ |
| | $(\frac{h}{8}, \frac{w}{8}, 16) \to (\frac{h}{4}, \frac{w}{4}, 8)$ | Deconv. | $4 \times 4$ | $2 \times 2$ |
| $g$ | $(\frac{h}{4}, \frac{w}{4}, 8) \to (\frac{h}{2}, \frac{w}{2}, 4)$ | Deconv. | $4 \times 4$ | $2 \times 2$ |
| | $(\frac{h}{2}, \frac{w}{2}, 4) \to (h, w, 13)$ | Deconv. | $4 \times 4$ | $2 \times 2$ |
| | $(h, w, 13) \to (\frac{h}{2}, \frac{w}{2}, 64)$ | Conv. | $4 \times 4$ | $2 \times 2$ |
| | $(\frac{h}{2}, \frac{w}{2}, 64) \to (\frac{h}{4}, \frac{w}{4}, 128)$ | Conv. | $4 \times 4$ | $2 \times 2$ |
| $d$ | $(\frac{h}{4}, \frac{w}{4}, 128) \to (\frac{h}{8}, \frac{w}{8}, 256)$ | Conv. | $4 \times 4$ | $2 \times 2$ |
| | $(\frac{h}{8}, \frac{w}{8}, 256) \to (1, 1, 1)$ | Conv. | $4 \times 4$ | $1 \times 1$ |

Table 1: Super Mario Bros Level Generation network architecture.

---

[*]Equal contributions.

[†]This work was partially carried out when PM was working at the University of Trento.

### 1.1.1 Details about the *pipes* constraint

Figure 1: A decomposed pipe

As reported in the main paper, experiment with the constraint on pipes has been run for $12000$ epochs. Figure 1 shows the various parts composing a pipe and their disposition. Suppose to call $\mathbf{a}$ the matrix boolean variables corresponding to the output of the generator $\mathbf{x}$ with shape $2 \times 2\times$. Remember that we apply the constraint separately to windows of size $2 \times 2$, with each pixel having $5$ channels. The four channels represent the probabilities of the tiles: [*top-left*, *top-right*, *body-left*, *body-right*, *others*]. In particular, in the last channel we collapse all the probabilities of the tiles that do not belong to pipes (air, monsters, walls, ...). Then, given the $2 \times 2 \times 5$ boolean vector, the list of the clauses composing the final constraint can be written as:

| | |
|---|---|
| $(\mathbf{a}_{0,0,0} \iff \mathbf{a}_{0,1,1})$ | top-left tile requires top-right tile on the right and vice-versa |
| $(\mathbf{a}_{0,0,2} \iff \mathbf{a}_{0,1,3})$ | body-left tile requires body-right tile on the right and vice-versa |
| $(\mathbf{a}_{0,0,0} \to \mathbf{a}_{1,0,2})$ | top-left tile requires body-left tile below |
| $(\mathbf{a}_{0,1,1} \to \mathbf{a}_{1,1,3})$ | top-right tile requires body-right tile below |
| $(\mathbf{a}_{1,0,2} \to (\mathbf{a}_{0,0,2} \vee \mathbf{a}_{0,0,0}))$ | body-left tile requires body-left of top-left above |
| $(\mathbf{a}_{1,1,3} \to (\mathbf{a}_{0,1,3} \vee \mathbf{a}_{0,1,1}))$ | body-right tile requires body-right of top-right above |
| $\displaystyle\bigwedge_{i=0}^{1} \bigwedge_{j=0}^{1} \mathrm{OHE}(\mathbf{a}_{i,j})$ | One hot encoding over all the 4 positions |

Notice that first two indexes describe the position, e.g. $0,0$ means the upper left corner of the $2 \times 2$ window, and the third index defines the tile type.

### 1.1.2 Details about the *reachability* constraint

The feedforward neural network $\phi$ is implemented using a CNN with two final dense layers, which architecture is described in Table 2.

| Input Shape $\to$ Output Shape | Layer Type | Kernel | Stride |
|---|---|---|---|
| $(h, w, 13) \to (h, w, 8)$ | Conv. | $3 \times 3$ | $1 \times 1$ |
| $(h, w, 8) \to (h, w, 16)$ | Conv. | $5 \times 5$ | $1 \times 1$ |
| $(h, w, 16) \to (h, w, 24)$ | Conv. | $7 \times 7$ | $1 \times 1$ |
| $(h, w, 24) \to (h, w, 32)$ | Conv. | $9 \times 9$ | $1 \times 1$ |
| $(h, w, 32) \to (h, w, 64)$ | Conv. | $11 \times 11$ | $1 \times 1$ |
| $(h, w, 64) \to (h, w, 96)$ | Conv. | $13 \times 13$ | $1 \times 1$ |
| $(h, w, 96) \to (h, w, 128)$ | Conv. | $15 \times 15$ | $1 \times 1$ |
| $(h, w, 128) \to (h, w, 192)$ | Conv. | $17 \times 17$ | $1 \times 1$ |
| $(h, w, 192) \to (h, w, 32)$ | Dense | - | - |
| $(h, w, 32) \to (h, w, 2)$ | Dense | - | - |

Table 2: Reachability Network architecture.

Performances of the approximation network $\phi$ include an accuracy and an F1-score higher than $94\%$. Picture 2 shows examples of how reachability maps have been approximated with $\phi$. The first column contains levels generated by the GAN. The binary maps have been obtained by summing the probabilities of all the solid tiles (ground, pipes, ...), given in white. The second column contains the reachability maps, computed by the *A\** agent and averaged over 100 different runs. Finally, the

Figure 2: Given some generated levels (first column), this picture compares real reachability maps computed by the *A\** agent (second column) and approximated by $\hat{P_R}$ (third column).

third column shows the reachability maps approximated by the $\phi$ neural network. Notice how well does $\phi$ work: the second and third columns are almost indistinguishable. $P_R$ and $\hat{P_R}$ are such that $A* : P_g \to P_R$ and $\phi : P_g \to \hat{P_R}$

## 1.2 Molecule Generation

The MolGAN architecture is composed of three networks, the generator $G$, the discriminator $D$ and the reward network $R$. $D$ and $R$ share the same architecture, but are trained with different objectives. $R$ is trained to predict the product of the QED, SA, logP metrics (in $[0, 1]$). $G$ is optimized to produce samples that maximize the output of $R$ and are convincing to $D$, on top of that, the conditional semantic loss is also applied.

While $R$ is trained in parallel with $D$ and $G$, the loss of $G$ with respect to the output of $R$ is activated only after 150 epochs, at which point the adversarial loss stops being used. The semantic loss is applied to $G$ from the start to the end of the training. The weight of the semantic loss is equal to $0.9$, whereas the adversarial loss of $G$ has a weight of $0.1$.

Improved WGAN is used as the adversarial loss between $G$ and $D$ ($n_{critic}$ = 5), whereas $R$ is trained to estimate the desired target by mean squared error, and $G$ is trained with deep deterministic policy gradient w.r.t. the output of $R$, which is seen as a reward to maximize.

Training proceeds until the uniqueness of the batch falling below $0.2$. During training, the learning rate is set at a constant value of $0.001$, the batch size is 32 and there is no dropout. Adam with $\beta_1 = 0.9$ and $\beta_2 = 0.999$ is the optimizer of choice. Batch discrimination is used.

Results are obtained by evaluating a batch of 5000 generated samples.

The input noise $z$ has dimension 32 if the semantic loss is not applied, 36 if it is applied; with the first 32 dimensions sampled from a standard normal distribution and the last four from a uniform distribution in $[0, 1]$.

The maximum number of nodes for each molecule is 9, with 5 possible atom types and 5 bond types. Each molecule is represented by two matrices, one mapping each node to a label, and one adjacency matrix informing about the presence or lack of edges between nodes, and their type.

The input noise is received by $G$ and processed by three fully connected layers of $128, 256, 512$ units each, while $\mathtt{tanh}$ acts as the activation function; a linear projection followed by a softmax is then applied to the output of the last layer to have it matching the size of the adjacency matrices.

$D$ and $R$ share the same architecture (no parameters are shared), based on two relational graph convolutions [3] of $64, 32$ hidden units, followed by an aggregation as in [3] to obtain a graph level representation of 128 features. Two fully connected layers of $128, 1$ units then reduce the graph embedding to a single output value, with $\mathtt{tanh}$ as the activation for the hidden layer and with sigmoid being applied on the output in the case of $R$. The MolGAN architecture is trained on the QM9 dataset [4, 5] composed of $5000$ training examples.