[Reviews · NeurIPS 2020]

Review 1

Summary and Contributions: The paper proposes a learning objective for incorporating constraints on the output space for generative modeling using GANs. To this end, the paper proposed to augment the GAN learning objective with a semantic labeling loss term that improves the optimization landscape. Empirical validation on a range of datasets supports the proposed method.

Strengths: + problem studied is of intellectual and practical interest to the community + incorporation of the semantic loss term is principled and algorithmically simple + empirical validation is well-explained, effective and comprehensive in most cases

Weaknesses: - a lot of theoretical exposition is trivial, verbose and unnecessary. In particular, Theorem 1, Corollary 1, Proposition 1 are all natural results that are immediate from the use of consistent loss terms. - the novel aspects of the work are not clearly explained. SL term is from prior work. Incorporating it within GANs however raises some challenges. For example, estimating SL term via Monte Carlo will give biased estimates. The authors allude to the use of knowledge compilation instead; it is however unclear the implications of this technique. Are the estimates biased or unbiased? Does the variance in estimation increase significantly in preferring the lower bound SL to the original loss term? Are there any approximation guarantees? - Is there any empirical evidence in support of conditional CANs? It seemed like an interesting and useful extension of CANs but unfortunately the exposition in the main paper is brief. - The formulation for f_ALT is the same as that of training GANs using integral probability metrics for different choice of discriminator families F. Should make that explicit and cite relevant works eg, MMD-GAN. - how is lambda varied over the course of training? a fixed value might be suboptimal and one would hope to anneal it during the course of training - in Table 3, can the authors shed light on why would the validity decrease with CANs? That seems to be the central USP of the proposed method.

Correctness: Yes, the claims and empirical validation are appropriate barring concerns already raised above. For rigor with GANs, it is highly desirable to repeat model training with atleast 5 seeds and report standard errors for all experiments.

Clarity: The paper is clearly written for most parts. Some of the decorative math (thm 1, cor 1, prop 1) could be omitted since they are trivial results and instead more real estate could be devoted to practical design choices relevant to the above algorithm. Typos/unclear: - (6) in L 110 should be \phi(x) - should the subscript in the prod terms be z_i?

Relation to Prior Work: To the best of my knowledge, yes. recently, there has been promising work in using resampling approaches (eg, [1, 2, 3]) for improving the sample quality and bias of generative models including GANs after training. In principle, while the implementation is very different in being applied post training, the approach mirrors the one adopted in CANs---use any default generator and correct for its mistakes with additional training using a relevant loss (eg, SL). Discussing how these approaches compare and complement with CANs would make for a more complete and richer discussion. References: [1] Boosted Generative Models Aditya Grover, Stefano Ermon AAAI 2018. [2] Discriminator Rejection Sampling Samaneh Azadi, Catherine Olsson, Trevor Darrell, Ian Goodfellow, Augustus Odena ICLR 2019. [3] Bias Correction of Learned Generative Models using Likelihood-free Importance Weighting Aditya Grover, Jiaming Song, Alekh Agarwal, Kenneth Tran, Ashish Kapoor, Eric Horvitz, Stefano Ermon NeurIPS 2019.

Reproducibility: Yes

Additional Feedback: See comments and questions above. ####### Post-rebuttal: Thanks for the rebuttal. It answers all but one question regarding the approximation quality and the bias/variance of the estimates obtained via KC. I am increasing my score to a 7.


Review 2

Summary and Contributions: This paper presents a new type of GAN, constrained adversarial networks (CAN), which use a semantic loss to force the samples to adhere to some desired constraints. They point out that GANs are inherently unlikely to generate valid images because of the curse of dimensionality and the fact that they faithfully reproduce noise. CANs are based on previous work which defines a semantic loss function capable of constraining networks to satisfy boolean constraints compiled as circuits. These circuits can be large but CANs don't need the compiled constraints at inference time. In some cases the constraints are altogether intractable. For these cases they use a net (trained on A* traces) to predict when the constraint is satisfied rather than explicitly encoding it.They point out that constraints can be added on any variables including latent variables, allowing them to use it to conditionally generate like InfoGANs. They evaluate on various game generation tasks as well as a molecule generation task, using a conditional CAN.

Strengths: -as far as I can tell this is a novel use of the semantic loss ideas in the GAN setting -GANs are of course a hot topic within ML, so this paper is quite relevant -the results look very convincing and inference time is roughly comparable (there is an increase in training time, but that is a sunk cost for the life of the model)

Weaknesses: -the paper is heavily dependent on the SL paper and perhaps you could say that it is not conceptually such a big leap to add SL to GANs -there is a very slight drop in validity when using CANs on the molecule experiment.

Correctness: The paper seems correct. I went through some of the proofs and they seemed ok to me.

Clarity: The paper is very well written, very clear, and a pleasure to read.

Relation to Prior Work: The prior work looks fine.

Reproducibility: Yes

Additional Feedback: I have read the rebuttal and stand by my original score.


Review 3

Summary and Contributions: The paper studies the problem of learning to generate structured objects (e.g., Super Mario Bros levels) using generative adversarial networks. The proposed approach augments the GAN objective with additional structured constraints which are based on previous work including semantic loss (ICML2018) and knowledge compilation (JAIR2002). The constraints are used to guide the GAN training, and can be discarded at inference time so that inference is fast without the need of (possibly costly) constraint evaluation. Experiments on Super Mario Bros level generation and Molecule generation are conducted.

Strengths: - The problem of integrating structured constraints with GANs (and more generally, deep neural networks) is of great practical importance in many real-world settings. - The adoption of semantic loss and knowledge compilation at training time to regularize the GAN training and discard the costly constraint evaluation at inference time looks reasonable and well-motivated. - Experiments on the Super Mario Bros level generation shows improvement over a couple of baselines.

Weaknesses: - The method section looks not self-contained and lacks descriptions of some key components. In particular: * What is Eq.(9) for? Why "the SL is the negative logarithm of a polynomial in \theta" -- where is the "negative logarithm" in Eq.(9)? * Eq.(9) is not practically tractable. It looks its practical implementation is discussed in the "Evaluating the Semantic Loss" part (L.140) which involves the Weighted Model Count (WMC) and knowledge compilation (KC). However, no details about KC are presented. Considering the importance of the component in the whole proposed approach, I feel it's very necessary to clearly present the details and make the approach self-contained. - The proposed approach essentially treats the structured constraints (a logical rule) as part of the discriminator that supervises the training of the generator. This idea looks not new -- one can simply treat the constraints as an energy function and plug it into energy-based GANs (https://arxiv.org/abs/1609.03126). Modeling structured constraints as a GAN discriminator to train the generative model has also been studied in [15] (which also discussed the relation b/w the structured approach with energy-based GANs). Though the authors derive the formula from a perspective of semantic loss, it's unclear what's the exact difference from the previous work? - The paper claims better results in the Molecule generation experiment (Table.3). However, it looks adding the proposed constrained method actually yields lower validity and diversity.

Correctness: The claim about better results in the Molecule generation experiment (Table.3) looks not correct, as the approach harms the validity and diversity.

Clarity: Descriptions of the key components in the proposed approach are missing. See above for more details

Relation to Prior Work: It's unclear what's the difference from previous frameworks such as energy-based GANs and [15]. See above for more details.

Reproducibility: No

Additional Feedback: The authors' feedback does address my concern about the empirical results (drop of validity). Also, the problem of incorporating structured knowledge with GANs is relevant and interesting. Thus I'd increase my rating to 6. My main concern is that, the use of KC for efficient constraint evaluation, as one of the major contributions (besides the other technical contribution of using SL for differentiable knowledge encoding), is not super clear to me. Also, the discussions of energy-based GAN and [15], should be made clearer.


Review 4

Summary and Contributions: This paper focuses on developing a generative model which produces outputs satisfying known expressible constraints. This is done using a GAN formulation with the addition of an additional penalty term, which the authors title "Conditional Adversarial Networks". The additional term, called semantic loss, adds a penalty based on the distribution of generated samples which violates the specified constraints. This term is implemented in practice by using Knowledge Compilation to build a circuit which allows for more efficient evaluation in some cases. Experiments demonstrate the ability of this approach to produce fewer (or similar amounts of) constraint-violating structures than other approaches while requiring less inference time.

Strengths: The application of the semantic loss (which was originally developed for discriminative semi-supervised learning) to generative modeling is a neat (and sensible) idea. The experiments indicate that the approach accomplishes what it sets out to do without increasing the time required to train an egregious amount; moreover, this technique has the benefit that it does not impact the time required for inference. In addition, as one of the experiments demonstrates, this development is orthogonal to some other approaches, and its strengths can be combined with others. There are many domains where outputs are required to satisfy constraints to be valid, and so any attempt at improving the ability of a generative model to do so will be beneficial to many.

Weaknesses: -Section 4.2 claims that this approach would be able to learn to generate structures that satisfy a set of constraints given a dataset that is "noisy" (i.e. contains structures that do not satisfy all of the constraints). This is a very interesting idea, but it is not expanded upon or evaluated in the experiments section, so it is unclear whether this would hold true in practice. -This technique can only be used if the constraints can be expressed using first-order logic (and which can be compiled to a sufficiently compact circuit). This may be challenging/intractable in practice. This concern can be bypassed in certain scenarios - e.g. if a model can be used to predict information out of which a constraint can be constructed, as in the "reachability" experiment - but this may be a challenging task for more general problems. - There are no guarantees on whether outputs of the model during inference satisfy constraints. This is only a concern if it is intractable to check whether an output is valid, however.

Correctness: The listed claims are correct as far as I am aware.

Clarity: The paper is easy to follow and explained in sufficient detail for the most part. I do think the subsection titled "The embedding function" starting on line 157 could be explained a bit more clearly, however; I only understood what this paragraph was trying to explain once I read the description of the model within the supplementary material.

Relation to Prior Work: This work is placed within the context of unconstrained GANs, older approaches that do not contain a representation learning component, and problem-specific approaches. The difference between CANs and these other approaches is made clear. Not many baselines are provided during experimentation, however.

Reproducibility: Yes

Additional Feedback: I think the work is interesting and I'm generally leaning towards acceptance. However, the experiments could use a few more points of comparison against prior techniques; this is especially true for molecule generation, which is widely studied at this point. Demonstrating that this approach can be applied to a wide variety of models (or outperform those it cannot be applied to) would strengthen this paper. ------------------------------------------------------------------------------------------- POST AUTHOR RESPONSE EDIT: Thanks for your response. Although I still feel more tasks and baselines would have strengthened the paper (and hence I have not increased my score), I still think this paper is interesting enough to be accepted.

[Author Response · NeurIPS 2020]

We thank all the reviewers for their careful reviews. All minor issues will be fixed in the text and more insight into knowledge compilation will be given. Detailed replies follow.

**Common concerns**. *Novelty wrt Xu et al 2018*: We: 1) Studied the limitations of GANs in constrained settings; 2) Showed how CANs and the SL follow naturally from introducing constraints into the discriminator of GANs; 3) Extended this setup to conditional CANs; 4) Provided a practical implementation with an effective approximate solution for constraints that cannot be evaluated efficiently; 5) Evaluated the utility of the proposed techniques on representative tasks of practical interest. None of these follow trivially from Xu et al. *Knowlege compilation*: Roughly speaking, KC leverages distributivity to rewrite the polynomial in Eq. 9 so to speed up evaluation. This is achieved by identifying shared sub-computations and compactly representing the factorized problem using a DAG. Target representations for the DAG (OBDDs, DNNFs, . . . ) differ in succinctness and enable different polytime (in the size of the DAG) operations. We reuse existing KC techniques, in particular by compiling the polynomial in Eq. 9 into a Sentential Decision Diagram (SDD) that enables efficient WMC and therefore exact probabilistic computation. *SL decreases the validity of molecules*: In molecule generation, validity is encouraged using the reward-based approach of MolGANs, not the SL. The SL is used instead to improve diversity (using cond. constraints that encourage generating different mixtures of chemical elements). As a result diversity increases ($91\%$ to $98.98\%$) and so does uniqueness ($2.4$ to $2.5$). Validity drops a little ($97.4\%$ to $96.6\%$) because the SL pushes a competing objective. This is expected. We will clarify this in the discussion.

**Reviewer 1**. *Unnecessary theory*: Cor. 1 and Prop. 1 serve to position CANs relative to GANs. We will trim the most obvious steps. *Evaluating the SL*: The SL measures the mass allocated by $P_g$ to configurations that violate the constraints and it can be evaluated *exactly* and *efficiently* using KC. No sampling is needed. The cost of KC, which is performed once before training, can be reduced by simplifying the space of binary variables using $\phi$. *Link to IPMs*: Thank you, we were not aware of this link. We will cite the relevant papers. *Evidence for conditional CANs*: The increase in diversity in molecule generation is obtained by turning on/off generation of specific atoms (e.g., chlorine) using cond. CANs. This serves as validation for the approach. We will add per-atom statistics in the Supp. Mat. *Schedule of $\lambda$*: In level generation, $\lambda$ is increased as explained in lines 236 and 263. In molecule generation, $\lambda = 0.9$ as the SL is not used for validity, so a constant value is enough in practice. Will report in the text. *5 repeats*: Good point. The values we reported are the average over $5$ runs for mario and $8$ for molecules. The unreported std. devs. are small, around 2–3% for avg. validity of 96.6%. We did not report std. devs. as we couldn't get them for CMA-ES due to time constraints. We will amend this. *Relation to extra papers [1]-[3]*: Thanks for pointing out the works on resampling strategies. We will mention them clarifying the differences wrt to CANs. In particular: CANs are not learned using maximum likelihood, so [1] does not apply; [3] assumes absolutely continuous distributions and corrects the predictions using pointwise estimates, while CANs focus on discrete probabilistic models and use the SL, which is neither an estimate nor pointwise. [2] is more closely related: in a sense, CAN generators act like proposal distributions for rejection sampling wrt the constraint, which is however given and fixed.

**Reviewer 2**. Please see the common concerns above.

**Reviewer 3**. *Why Eq. 9?*: Eq 9 shows that $P_g(\psi)$ is a polynomial and thus differentiable, enabling end-to-end training. The negative logarithm appears in Eq. 7, we amended the text. *Relation to EBGANs*: True, constraints can be introduced as energy functions into EBGANs, but they must be made differentiable first. There are many ways to do so (e.g., using fuzzy logic, as discussed in lines 133–140). However, only the SL is probabilistically sound and semantic (i.e., its value does not depend on the encoding of the constraint, see Figure 1). Also, the SL can be evaluated exactly; no sampling is required. *Relation to [15]*: This work is complementary to ours as it implements a constraint learning component (using inverse reinforcement learning) rather than a constraint enforcing loss term, and it is useful when constraints are unknown or cannot be explicitly encoded (see our discussion in related work). Indeed we showed in the molecule generation experiment how to combine the advantages of constraint learning (using reinforcement learning) with constraint enforcement.

**Reviewer 4**. *Learning from invalid examples*: Good point. Experiments with noisy data will be carried out in a future study. These are not strictly necessary to highlight the benefits of CANs. *Non-approximable constraints*: Correct, some constraints are too hard for both KC and for neural net approximation. However, this issue is not specific to CANs and indeed it affects all approaches to learning under constraints (generative or not). *Generated objects must be filtered*: Precisely. So long as the constraints are Boolean (or discrete), checking validity of a configuration is polynomial, so this is not an issue. We will clarify this. *Embedding function is unclear*: The embedding function $\phi$ extracts Boolean features/variables to which the constraints are then applied. In many cases $\phi$ is the identity map. We will clarify this. *More baselines*: For level generation, we compared against GAN + CMA-ES which was the SotA when we ran the experiments (and until very recently). MolGAN is also a very respectable method. Still, our goal is not to challenge the SotA but rather to show that CANs lead to clear improvements in many settings and for many tasks (e.g., increasing validity, improving diversity). Also, we did not tweak CANs for specific domains, although this would improve performance. We defer a detailed study of CANs to specific domains (especially level design) to future work.

[Meta-Review · NeurIPS 2020]

This work aims at estimating generative distributions of structured objects that satisfy certain semantic constraints (in first-order logic). The authors achieve this goal by adding a "semantic loss" to the GAN’s learning objective and using Knowledge compilation (KC) to build a circuit that allows efficient evaluation. Experiments on game-level generation tasks and a molecule generation task support the proposed method. Strengths: i) Incorporating structured constraints in GAN models is both intellectually and practically interesting; ii) The experiments are comprehensive and convincing in most cases; and iii) the paper is clearly written for most parts. Weaknesses: i) The main novelty and similarities/differences/limitations in comparison with other related approaches (e.g., Xu 2018) are not clearly discussed; and ii) technical descriptions for the KC techniques should be made clearer and self-contained. The paper is recommended for acceptance. The authors are urged to inculcate reviewers' suggestions and address the outlined weaknesses in their final draft. Jingyi Xu, et al. "A Semantic Loss Function for Deep Learning with Symbolic Knowledge." International Conference on Machine Learning. 2018.